# Diversity of Bioactive Compounds in Microalgae: Key Classes and Functional Applications

**DOI:** 10.3390/md23060222

**Published:** 2025-05-22

**Authors:** Maslin Osathanunkul, Suebsuya Thanaporn, Lefkothea Karapetsi, Georgia Maria Nteve, Emmanouil Pratsinakis, Eleni Stefanidou, Giorgos Lagiotis, Eleni Avramidou, Lydia Zorxzobokou, Georgia Tsintzou, Artemis Athanasiou, Sofia Mpelai, Constantinos Constandinidis, Panagiota Pantiora, Marián Merino, José Luis Mullor, Luka Dobrovic, Leonardo Cerasino, Tomohisa Ogawa, Meropi Tsaousi, Alexandre M. C. Rodrigues, Helena Cardoso, Rita Pires, Daniel Figueiredo, Inês F. Costa, Catarina Anjos, Nikolaos E. Labrou, Panagiotis Madesis

**Affiliations:** 1Department of Biology, Faculty of Science, Chiang Mai University, Chiang Mai 50200, Thailand; maslin.cmu@gmail.com (M.O.); thanapornsuebsuya@gmail.com (S.T.); 2Institute of Applied Biosciences, Centre for Research and Technology Hellas, 57001 Thessaloniki, Greece; lefki8@certh.gr (L.K.); margodeves@certh.gr (G.M.N.); epratsina@certh.gr (E.P.); elenuba@certh.gr (E.S.); glagiotis@certh.gr (G.L.); leleav_av@certh.gr (E.A.); lzorzobokou@certh.gr (L.Z.); 3Laboratory of Molecular Biology of Plants, School of Agricultural Sciences, University of Thessaly, 38221 Volos, Greece; gtsintzou@uth.gr (G.T.); aathanasiou@certh.gr (A.A.); bsofia@uth.gr (S.M.); 4Laboratory of Enzyme Technology, Department of Biotechnology, School of Applied Biology and Biotechnology, Agricultural University of Athens, 75 Iera Odos Street, 11855 Athens, Greece; constantinos.c98@gmail.com (C.C.); pantiora@aua.gr (P.P.); 5Bionos Biotech S.L., Av. Fernando Abril Martorell, 106, 46026 Valencia, Spain; mmerino@bionos.es (M.M.); jlmullor@bionos.es (J.L.M.); 6Particula Group Ltd., Tina Ujevića 9, 51000 Rijeka, Croatia; luka.dobrovic@particula-group.com; 7IASMA Research and Innovation Centre, Fondazione Edmund Mach, via E. Mach, 1, 38010 San Michele all’Adige, TN, Italy; leonardo.cerasino@fmach.it; 8Laboratory of Enzymology, Graduate School of Agricultural Science, Tohoku University, 468-1 AzaAoba Aramaki, Aoba-ku, Sendai 980-8572, Japan; tomohisa.ogawa.c3@tohoku.ac.jp; 9Fresh Formula Private Limited Cosmetics Manufacturing Company, 1st km Lavriou Ave Koropiou—Markopoulou, 19400 Koropi, Greece; m.tsaoussis@freshline.gr; 10Necton S.A., Belamandil, 8700-152 Olhão, Portugal; alexandre.rodrigues@necton.pt (A.M.C.R.); ines.costa@necton.pt (I.F.C.); catarina.anjos@necton.pt (C.A.); 11Allmicroalgae-Natural Products S.A., Pataias-Gare, 2445-413 Pataias, Portugal; helena.cardoso@allmicroalgae.com (H.C.); ri.pires@campus.fct.unl.pt (R.P.); 12LAQV-REQUIMTE, Department of Chemistry, Nova School of Science and Technology, Nova University Lisbon, Campus da Caparica, 2829-516 Caparica, Portugal; 13GreenCoLab—Associação Oceano Verde, 8005-139 Faro, Portugal; daniel.figueiredo@tecnico.ulisboa.pt; 14Centro de Ciências Do Mar Do Algarve, Universidade Do Algarve Campus Gambelas, 8005-139 Faro, Portugal

**Keywords:** biomass, carbon dioxide utilization, cosmeceuticals, food supplements, pharmaceuticals

## Abstract

Microalgae offer a sustainable and versatile source of bioactive compounds. Their rapid growth, efficient CO_2_ utilization, and adaptability make them a promising alternative to traditional production methods. Key compounds, such as proteins, polyunsaturated fatty acids (PUFAs), polyphenols, phytosterols, pigments, and mycosporine-like amino acids (MAAs), hold significant commercial value and are widely utilized in food, nutraceuticals, cosmetics, and pharmaceuticals, driving innovation across multiple industries. Their antiviral and enzyme-producing capabilities further enhance industrial and medical applications. Additionally, microalgae-based biostimulants and plant elicitor peptides (PEPs) contribute to sustainable agriculture by enhancing plant growth and resilience to environmental stressors. The GRAS status of several species facilitates market integration, but challenges in scaling and cost reduction remain. Advances in biotechnology and metabolic engineering will optimize production, driving growth in the global microalgae industry. With increasing consumer demand for natural, eco-friendly products, microalgae will play a vital role in health, food security, and environmental sustainability.

## 1. Introduction

Microalgae have emerged as a promising and sustainable alternative for the production of biomass and bioactive compounds, gathering increasing attention across various industries [1]. These organisms can thrive in a variety of water sources, ranging from freshwater to wastewater, providing versatile options for cultivation [2]. Additionally, the application of microalgae biotechnology is expanding commercially, with significant relevance in sectors such as food, pharmaceuticals, nutraceuticals, cosmetics, animal feed, and energy [3,4,5,6,7,8].

Microalgae, as photosynthetic microorganisms, can be cultivated under controlled conditions to yield high amounts of biomass enriched with valuable compounds such as lipids, proteins, carbohydrates, and pigments. The biomass production of microalgae is an emerging field with significant potential for various industrial applications [9,10]. Their ability to efficiently convert CO_2_ into organic matter through photosynthesis makes them particularly attractive for sustainable production practices [11,12,13] (Figure 1). Microalgae grow quickly, require minimal space, can grow on non-arable land, and can be cultivated in a variety of environments, including wastewater. They are considered a green source of bioactive compounds, offering an environmentally friendly alternative to traditional methods of industrial production, which often rely on resource-intensive processes. It is important to emphasize that both the microalgal species and the specific growth conditions critically influence the production and accumulation of high-value bioactive compounds. Notably, modulation of growth conditions can significantly impact the accumulation of these compounds [14], particularly through variations in light intensity, medium pH, temperature, and nutrient composition of the culture medium [15,16] (Table 1).

In the context of food applications, the “Generally Recognized As Safe” (GRAS) designation by the U.S. Food and Drug Administration (FDA) is a critical criterion for evaluating the safety of substances intended for use in food products. Several microalgal species have been granted GRAS status (Table 2), thereby permitting their incorporation as food ingredients in various formulations [17,18,19]. This regulatory approval is essential for facilitating the commercial use of microalgae in the food and nutraceutical sectors. This designation ensures that the organisms are deemed safe for consumption, minimizing the need for extensive purification processes when used in food additives. In contrast, if an organism lacks GRAS status, its derived ingredients must undergo stringent purification to meet safety standards, which can increase production costs. The GRAS status of certain microalgae helps to promote consumer confidence, aids food manufacturers in regulatory compliance, and supports the integration of microalgae into a broader array of food and beverage products [17,18,19]. This contributes to the growth of the functional foods market and supports the development of sustainable, nutrient-rich food sources [20,21]. A list of related products approved by the European Union is available at the European Algae Biomass Association website [22], while the European Food Safety Authority (EFSA) is also currently evaluating the safety of various microalgae-derived products.

High added-value compounds from microalgae (Figure 2) refer to substances with significant economic and industrial potential [23,24]. These compounds include proteins, lipids, carbohydrates, vitamins, and antioxidants, and several bioactive compounds including polyphenols, phytosterols, pigments, and mycosporine-like amino acids (MAAs) [24,25]. These compounds are valuable due to their diverse applications in industries like food, pharmaceuticals, and cosmetics. In addition, other emerging compounds currently under intense research include biostimulants, which enhance plant growth; plant elicitor peptides (PEPs), which activate plant defense mechanisms; antivirals with the potential to inhibit virus activity; and industrial and analytical enzymes, which play critical roles in manufacturing and research [26]. These compounds are anticipated to drive substantial growth in the global microalgae biotechnology market over the next decade, driven by consumer demand for high-quality, eco-friendly, and cost-effective products. This review highlights the increasing prominence of microalgae-derived compounds across agricultural and pharmaceutical applications, underscoring their versatile potential in meeting the needs of diverse industries.

**Table 1 marinedrugs-23-00222-t001:** Examples of growth conditions influencing microalgae production.

Species	Conditions				
	Illumination Period	Temperature (°C)	Nutrient Medium	pH	References
*Chlorella vulgaris*	24 h light/day	25	BG11 medium	9	Deniz, 2020 [27]
*Limnospira platensis* *	12 h light/day	32	Zarrouk’s medium	9.5	Soni et al., 2019 [28]
*Dunaliella salina*	15 h light/day	25	f/2 medium	7.2	Al-Mhanna et al., 2023 [29] Borovkov et al., 2020 [30]
*Nannochloropsis oculata*	16 h light/day	25	f/2 medium	7.2	Maglie et al., 2021 [31]
*Haematococcus lacustris*	16 h light/day	30	Bold’s basal medium	5	Hanan et al., 2013 [32] Nahidian et al., 2018 [33]

* Formerly known as Arthrospira platensis or Spirulina platensis.

**Table 2 marinedrugs-23-00222-t002:** Selected microalgal species that have received GRAS (Generally Recognized As Safe) status from the U.S. Food and Drug Administration (FDA), including notification numbers, date of closure, and their representative food applications.

Substance/Microalgal Species	Date of FDA Closure	GRAS Notification No.	Common Applications
*Limnospira platensis*	27 July 2011	GRN 391	Dietary supplements, protein fortification, natural colorant
*Chlorella vulgaris*	8 January 2007/23 September 2016	GRN 224/641	Smoothies, nutrition bars, fortified beverages
*Dunaliella salina*	21 May 2002	GRN 103	Natural β-carotene source, food colorant, functional ingredient
*Haematococcus lacustris*	20 October 2010/11 June 2013	GRN 314/452	Astaxanthin-rich supplements and functional foods
*Schizochytrium* sp.	17 February 2004/14 May 2018	GRN 137/778	DHA omega-3 for infant formula and nutrition products
*Euglena gracilis* (dried biomass)	19 July 2017	GRN 697	β-1,3-glucans; used in beverages, supplements, and snacks
*Euglena gracilis*	22 May 2017	GRN 686	β-1,3-glucans; used in beverages, supplements, and snacks
*Aurantiochytrium limacinum* TKD-1 (algal oil, ≥45% DHA)	25 February 2022	GRN 1008	DHA-rich algal oil for food and supplements
*Chlorella protothecoides* strain S106	12 Jun 2014	GRN 519	High-protein flour (40–75%), used in fortified foods
*Chlorella sorokiniana*	21 December 2021	GRN 986	Powder and micro powder for dietary applications
*Dunaliella bardawil*	19 August 2010	GRN 351	β-carotene-rich natural food colorant
*Haematococcus lacustris*	4 June 2015	GRN 580	Astaxanthin esters for supplements and fortified foods
*Ulkenia* sp. SAM2179 (micro-algal oil)	21 January 2010	GRN 319	DHA-rich oil used in nutritional applications

## 2. High Added-Value Compounds from Microalgae

### 2.1. Proteins

Proteins derived from microalgae are of particular interest due to their high nutritional value, containing essential amino acids necessary for human health and wellbeing [34,35,36,37]. Moreover, microalgae offer a sustainable source of proteins that can be produced efficiently without competing with food crops for arable land. Advances in biotechnological techniques have enabled the optimization of microalgae strains to enhance protein content and productivity, paving the way for their use in food supplements, animal feeds, and biomedical applications [25,38].

The protein content of microalgae offers a compelling alternative to traditional protein sources, both nutritionally and functionally. For example, proteins derived from Chlorella, Limnospira, and *Dunaliella salina* contain all essential amino acids required by the human body, making them comparable in nutritional value to animal-based protein sources such as meat, dairy, and eggs [39,40,41]. This makes them particularly valuable for vegetarian and vegan diets, where complete protein sources are often limited.

Microalgal protein concentrates are primarily utilized as nutraceuticals or added to functional food recipes [39,42,43,44]. In addition, microalgal proteins have been increasingly incorporated into the cosmetics industry, particularly in skincare products. Notably, proteins from the genera Limnospira and Chlorella are among the most commonly utilized sources [13,25,45,46].

### 2.2. Polyunsaturated Fatty Acids (PUFAs)

Polyunsaturated fatty acids (PUFAs), including arachidonic acid (AA), docosahexaenoic acid (DHA), eicosapentaenoic acid (EPA), and γ-linolenic acid (GLA), are essential fatty acids that support cardiovascular health, cognitive development, and immune function [47]. As the human body cannot synthesize these fatty acids, they must be obtained through diet, making external sources vital for human health and nutrition.

Microalgae are recognized as a sustainable and efficient source of PUFAs, particularly as alternatives to fish-derived oils. Several species have demonstrated high PUFA production capacity, including *Chaetoceros calcitrans*, *Monodus subterraneus*, *Phaeodactylum tricornutum*, *Porphyridium cruentum*, *Nannochloropsis* spp., *Crypthecodinium cohnii*, *Tisochrysis lutea*, *Pavlova gyrans*, and *Pavlova* sp. [48,49]. These microalgae produce a diverse array of omega-3 and omega-6 fatty acids and are increasingly utilized in functional foods and nutraceutical formulations aimed at delivering anti-inflammatory and cardioprotective effects. The green microalga *Lobosphaera incisa* is an oleaginous eukaryotic alga that is rich in AA (20:4) [50].

Limnospira contributes to this group as a notable source of GLA, a rare omega-6 fatty acid with known health-promoting properties [47]. While its total PUFA content may be lower, compared to some marine microalgae, Limnospira’s accessibility, nutritional profile, and regulatory approval for human consumption have made it a popular ingredient in supplements and fortified products. Collectively, PUFA-rich microalgae offer promising applications in the nutraceutical sector, providing plant-based, environmentally friendly sources of essential fatty acids to meet growing consumer demand for sustainable health solutions.

### 2.3. Polyphenols

To counteract oxidative stress arising from both endogenous and environmental sources, microalgae produce a diverse range of high- and low-molecular-weight antioxidant molecules. Among these are polyphenols, a broad class of secondary metabolites that includes flavonoids, isoflavonoids, phenolic acids, lignans, and stilbenes [51,52,53,54,55]. The primary benefit of polyphenolic compounds for human health lies in their potent antioxidant activity. Phenolic compounds enhance cellular antioxidant capacity by upregulating enzymes involved in oxygen metabolism and xenobiotic detoxification, while simultaneously downregulating signaling pathways associated with inflammation. Through these mechanisms, they effectively scavenge and reduce intracellular reactive oxygen species [56,57]. Flavonoids, a major subclass of polyphenols, exhibit a range of additional health-promoting effects. Compounds such as luteolin, quercetin, and rutin are particularly notable for their potential in treating cardiovascular disorders, largely due to the presence of a catechol group in the B-ring and an unsaturated C2–C3 bond in conjunction with a 4-oxo function—structural features associated with strong antioxidant and vascular-protective activities. The angiotensin-converting enzyme (ACE), a target of modern medications for the treatment of arterial hypertension, is naturally inhibited by other flavonoids such as kaempferol and apigenin. Moreover, some polyphenols have antiviral activities [53]. The primary mechanism for suppressing viral replication, mostly of ssRNA-positive viruses, is polyphenols’ capacity to prevent reactive oxygen species from spreading within cell systems [58]. In the instance of coronavirus infection (SARS-CoV-2), apigenin and kaempferol block the renin–angiotensin–aldosterone system (RAAS), which contributes to virus entrance into lung cells [53]. Another group called phlorotannins can prevent the human immunodeficiency virus (HIV) from replicating [59]. In the context of cosmetic applications, microalgal extracts rich in polyphenols are already utilized in products designed to promote hair growth, protect against UV-induced damage, regulate skin pigmentation, enhance skin firmness, and combat signs of aging [60]. In addition to their health-promoting properties, polyphenols, when incorporated into cosmetic and nutraceutical formulations, also function as effective natural preservatives. Their antioxidant activity helps extend the shelf life of foods and beverages by enhancing overall antioxidant capacity and preventing oxidation-induced degradation [57,61]. Limnospira species have also been shown to be rich in antioxidant compounds, such as phenolic acids [62,63,64]. Additionally, the nitrogen-fixing genera Nostoc and Anabaena are of particular interest due to their notable antiviral, antioxidant, and anticancer activities [65,66,67].

### 2.4. Phytosterols

More than 100 different types of phytosterols have been identified so far [68] (Figure 3). The phytosterols used in industry mainly originate from plants [69]; however, microalgae are also a valuable source. Various classes of algae—including Chlorophyceae, Phaeophyceae, and Rhodophyceae—have been reported to produce phytosterols [70]. For instance, species such as *Nannochloropsis gaditana*, *Phaeodactylum tricornutum*, *Nannochloropsis* sp., and *Tisochrysis lutea* have been shown to contain phytosterols ranging from 7 to 34 g/kg dry weight (0.7–3.4%) [71]. Additionally, *Tetraselmis* sp. M8, *Pavlova lutheri*, and *Nannochloropsis* sp. BR2 have demonstrated phytosterol contents between 0.4% and 2.6% of dry weight. *P. lutheri* has been reported to reach up to 5.1%, depending on factors such as cultivation medium, salinity, and growth duration [71,72,73].

Several studies have reported that dietary phytosterol intake can interfere with the absorption of fat-soluble nutrients such as β-carotene and vitamin E [74]. As a result, it is often recommended to increase the intake of these micronutrients concurrently to compensate for the reduced absorption caused by phytosterols [75]. However, microalgae present a promising alternative to multiple supplementations, as they can naturally provide high levels of essential nutrients, including vitamins, phytosterols, and antioxidants, potentially fulfilling a substantial portion of human dietary requirements. However, microalgae could substitute the multi-supplements, since they could cover most human dietary needs in vitamins, phytosterol content, and antioxidants.

Phytosterols have been widely recognized for their beneficial effects on human health [72,76]. Regular consumption of 2–3 g per day has been shown to effectively lower low-density lipoprotein (LDL) cholesterol levels, thereby supporting cardiovascular health. In addition to their lipid-lowering properties, phytosterols also exhibit anti-inflammatory, anti-atherogenic, and anticancer activities [77]. Notably, ergosterol, a phytosterol found in microalgae, has demonstrated significant cytostatic effects against human colorectal adenocarcinoma cells [78]. Furthermore, fucosterol and its oxygenated derivatives, isolated from the brown alga *Sargassum carpophyllum*, have shown cytotoxic activity against various cancer cell lines [79].

### 2.5. Pigments

The primary pigments in microalgae are classified as chlorophylls, carotenoids, and phycobilins (Figure 4). These compounds have been shown in recent years to exhibit valuable health-promoting properties, including antioxidant activity, provitamin functionality, immune modulation, and anti-inflammatory effects [80,81]. Consequently, they are primarily utilized in the food, pharmaceutical, and cosmetic industries as natural colorants, dietary supplements, or sources of bioactivity [82].

#### 2.5.1. Chlorophylls

Chlorophylls (Figure 4a) are found in all photoautotrophic organisms and are necessary for photosynthesis [83,84]. Chlorophylls are gaining increasing importance as natural colorants in the food, pharmaceutical, and cosmetic industries due to their intense green pigmentation and the rising consumer preference for naturally derived ingredients [85,86]. Microalgae of the genus *Chlorella*, whose chlorophyll content represents about 7% of its biomass (five times more than that of Limnospira [42]), are widely used as chlorophyll producers.

#### 2.5.2. Carotenoids

Carotenoids are another class of pigments that are abundantly found in microalgae. The primary characteristics of these vividly colored molecules, which range in hue from yellow to red, are their antioxidant capabilities and dyeing ability [85,86,87]. Carotenoids are important pigments that play a significant role in photosynthesis, having a double function, in light harvesting and in protecting the photosynthetic apparatus from excess energy [88,89]. These carotenoids are characterized as primary carotenoids and are essential for cellular survival. On the other hand, secondary carotenoids are not necessary for photosynthesis [90] but they are accumulated in stressful conditions like high light, salinity, and nutrient starvation [91,92].

The biosynthesis of carotenoids in photosynthetic microalgae starts from isopentenyl pyrophosphate (IPP) or dimethylallyl diphosphate (DMAPP). In the presence of desaturase, ζ-carotene can be formed in algae or higher plants (the metabolic pathways in bacteria or fungi would slightly differ). Astaxanthin, an economically important carotenoid, is usually synthesized from canthaxanthin or zeaxanthin by photosynthetic microalgae [93]. Although carotenoids can be synthesized chemically at a lower cost, their use has been associated with potential health risks, including an increased incidence of certain diseases such as lung cancer and cardiovascular diseases [94,95]. Thus, safety concerns lead to the increased market of the natural pigments, especially for human consumption [96]. Nevertheless, the cost of naturally produced carotenoids using microalgae is still questionable.

*Haematococcus lacustris*, *Dunaliella salina*, *Chromochloris zofingiensis*, and *Chlorella vulgaris* are the main species initially considered for the commercial production of carotenoids in large-scale cultures [90,97]. Many additional microalgal species are capable of producing and accumulating secondary carotenoids, particularly when cultivated under stress conditions such as high light intensity, nitrogen limitation, or salt stress [90]. Notable examples include *Auxenochlorella protothecoides* [98] and *Scenedesmus almeriensis* [99]. β-carotene is the first high-value product to be manufactured commercially from microalgae (Figure 4b) [100]. According to Bhalamurugan et al. (2018) [49] and Khanra et al. (2018) [42], *Scenedesmus almeriensis* and *Dunaliella tertiolecta* are notable microalgal producers of β-carotene. However, the most abundant and commercially significant natural source of β-carotene remains *Dunaliella salina* [100,101].

Microalgae-derived carotenoids have a wide range of applications across multiple industries. In the food industry, they are commonly used as natural additives for coloring and for their strong antioxidant properties [102]. For example, astaxanthin is employed as a natural antioxidant that can prevent lipid oxidation and extend the shelf life of food products. *Haematococcus lacustris’* ability to produce over 50 mg/g of astaxanthin makes it a significant natural source of this valuable carotenoid.

Another important area of application for carotenoids is the pharmaceutical industry, where they are valued for their therapeutic properties. Several carotenoids have been shown to offer protective effects against chronic diseases such as diabetes and cancer. For instance, β-carotene, fucoxanthin, and astaxanthin exhibit notable anticancer activity. Notably, Palozza et al. (2005) [103] reported that β-carotene inhibits the proliferation of human colon cancer cell lines. Fucoxanthin is reported to be a promising anticancer carotenoid that inhibits the growth of SK-hepatoma, leukemia cells, and prostate cancer [104]. Astaxanthin has been shown to play a protective role in cardiovascular health, primarily due to its potent antioxidant activity and its ability to reduce inflammation and lower LDL cholesterol levels [105,106]. In addition to their pharmaceutical benefits, carotenoids are widely used in the cosmetics industry for their antioxidant properties and anti-UV protective effects. These compounds are also believed to promote skin health by enhancing elasticity, improving hydration, and refining texture [107,108,109,110].

#### 2.5.3. Phycobiliproteins

Phycobiliproteins (PBPs), are a family of natural pigment–proteins from microalgae containing phycobilins (Figure 4c–f) [111]. PBPs are heterodimeric proteins composed of α and β subunits, with the β subunits being slightly larger than their α counterparts. Three (αβ) monomers assemble into (αβ) trimers, forming a circular disk-like structure (Figure 5). These trimers can further associate to create (αβ) hexamers. Each PBP subunit carries one to three linear tetrapyrrole chromophores, known as phycobilins, which are covalently bound to specific cysteine residues. The absorption properties of PBPs are determined by the type of phycobilins they contain (Figure 4c–f), including phycocyanobilin (PCB, λ_max_ = 640 nm), phycoerythrobilin (PEB, λ_max_ = 550 nm), phycourobilin (PUB, λ_max_ = 490 nm), and phycoviolobilin (PVB, λ_max_ = 590 nm) (λ_max_ was determined in distilled water). Based on their spectral characteristics, PBPs are categorized into three primary groups, allophycocyanin (APC), phycocyanin (PC), and phycoerythrin (PE), with additional subtypes distinguished by their unique absorption spectra.

Various microalgal species, such as *Limnospira platensis*, *Porphyridium* sp., *Aphanizomenon gracile*, *Neopyropia yezoensis*, the red alga *Porphyridium purpureum*, and several cryptophytes, are known to accumulate different levels of phycobiliproteins (PBPs), depending on cultivation conditions. For example, the *Nostoc* sp. SW02 strain has been reported to achieve a high PBP content (31.9%) when cultured under low light intensity and nutrient-limited conditions [111]. Industrial production of these pigments involves the utilization of the species *Porphyridium* sp., *Limnospira* sp., and *Aphanizomenon flosaquae* [48,49,80,81,85,86].

Phycobiliproteins (PBPs) are of significant commercial interest, not only as natural colorants but also for their potential pharmaceutical applications. PBPs have demonstrated a range of therapeutic properties, including antitumor, antioxidant, hepatoprotective, and neuroprotective effects. Among them, phycocyanin (PC), a well-studied member of the PBP family, has shown the ability to selectively induce apoptosis in cancer cells without harming normal cells [112]. Furthermore, PBPs have been reported to induce cell cycle arrest in various cancer types, including liver cancer [113], breast cancer [114], leukemia [115], lung cancer [116], and bone marrow cancer [117].

PBPs are highly used as colorants in the food industry due to their nontoxic nature. PC, extracted from *L. platensis*, when added to ice-cream can maintain its color for up to 182 days [118], and when added to yogurt, it increases color stability, viscosity, and, due to its antibacterial activity, extends the yogurt’s life [119]. Phycobiliproteins combined with skimmed milk result in a mixture of enhanced chemical stability and antioxidant capacity, which can be considered as an innovative beverage [120].

PBPs are also used as natural colorants in the cosmetics industry, in products like creams, lipsticks, and eyeliners [120]. Novel cosmetics using PBPs are also under development.

One main role of PBPs is to scavenge free radicals; thus, PBPs play an important protective role in different organs like the liver and kidneys and systems like the nervous and cardiovascular systems [121]. PBPs have also been found to play a protective role and improve Alzheimer’s disease symptoms, as they reduce the accumulation of amyloid β plaques in the brain [122].

PBPs are widely utilized as analytical reagents in molecular biology laboratories and serve as markers in various immunological techniques, including flow cytometry, microscopy, and DNA assays. Their strong and highly sensitive fluorescent properties make them particularly valuable for these applications [43,81,85,86,123,124].

### 2.6. Biostimulants

Plant development can be enhanced by the application of microalgae, cyanobacteria, or their formulations, such as biomass, extracts, or hydrolysates. The effectiveness of these biostimulants depends on the species of microalgae and the origin of the formulation used [6,125,126,127]. Furthermore, the combination of biostimulants with a specific plant species and the growing conditions play an equally important role. In addition, both the developmental stage and the environmental conditions at the time of the biostimulant application can directly influence the success of the biostimulant treatment. Biostimulant application has a direct impact on plant growth rate and yield, which is important especially for leafy vegetables like lettuce or spinach and herbs like oregano and basil, while it has been shown that it enhances their development under abiotic stress conditions. Biostimulants trigger nitrogen and carbon metabolism, resulting in enhanced growth. At the same time the carbohydrate, leaf, protein, and photosynthetic pigment (chlorophyll and carotenoid) content is increased [6,127]. Moreover, the application of biostimulants increases the metabolite content, which in turn improves the quality of the products and extents their shelf life. Biostimulants also enhance germination rates, resulting in early flowering and increased numbers of flowers. Finally, polysaccharide extracts from microalgae species like *Chlorella sorokiniana*, *Chlamydomonas reinhardtii*, *Porphirydium* spp., and *Dunaliella salina* enhance the expression of antioxidant enzymes like superoxide dismutase, peroxidase, and catalase in tomato plants under salt stress and thus can serve as biostimulants [128]. Additionally, green microalgae have the ability to produce lactic acid, a chiral organic acid that is essential for plant growth and development [126].

### 2.7. Plant Elicitor Peptides (PEPs)

Plant elicitor peptides (PEPs) constitute a novel tool for sustainable agriculture. As endogenous elicitors, PEPs contribute to the defense mechanisms of plants against bacterial, fungal, and herbivorous threats [129]. In agricultural applications, microalgae can function as stimulants for plant defense systems. These organisms release chemical compounds and structural components that induce a plant’s defensive responses. When applied directly to crop roots, microalgae activate systemic defense responses in plants, as the interactions between microalgae and plants are bidirectional. This phenomenon has led to the identification of priming-type responses in various crops, although the specific microalgae and molecules responsible remain unidentified. For instance, when broccoli and guar plant roots were inoculated with *Chlorella vulgaris*, it resulted in increased activity of antioxidant enzymes, including APX, CAT, SOD, and glutathione reductase. Furthermore, this inoculation induced an accumulation of flavonoid and phenolic compounds [130]. The elicitation technique has practical applications in the industrial sector, particularly in the production of plant-cell-derived chemicals. Plants employ defense mechanisms similar to priming to prepare for potential pathogen invasions. In cucumber plants, the foliar inoculation of *Desmodesmus abundans* triggers Systemic Acquired Resistance (SAR) and induces various cellular changes, which help mitigate the effects of *Colletotrichum orbiculare* disease. These cellular modifications include the accumulation of vesicles, the formation of sheaths around penetration hyphae, and the thickening of cell walls adjacent to intracellular hyphae [130]. Research conducted by Rachidi et al. in 2021 [131] has demonstrated that polysaccharides extracted from various microalgae species, including Chlorella and Dunaliella (members of Chlorophyta), as well as Porphyridium (belonging to Rhodophyta), function as effective plant elicitors in tomatoes. This elicitation results in enhanced activity of defense-related enzymes and increased synthesis of protective compounds, such as polyphenols and steroidal glycoalkaloids [131]. The exopolysaccharides produced by *Porphyridium sordidum* stimulate the expression of SA-related genes and PAL activity in *Arabidopsis thaliana* leaves, leading to decreased incidence of diseases caused by *Fusarium oxysporum* [132]. Additionally, green microalgae have the ability to produce lactic acid, a chiral organic acid that is essential for plant growth and development [133]. Poveda et al. (2022) have investigated the possibility of microalgae elicitors passing on plant defense activation to future generations. They cultivated tomato, pepper, and eggplant in vermicompost containing microalgae species *Ulothrix* spp. (*Chlorophyta*) and *Navicula* spp. (*Ochrophyta*). The seeds collected from these plants were then germinated in the presence of the pathogenic oomycete *Pythium* sp. Results showed a 90% increase in seedling survival, indicating a potential inherited resistance [134]. Rao et al. (2001) studied the effects of two microalgal species, *Haematococcus lacustris* and *Limnospira platensis*, used as elicitor treatments, on the accumulation of betalaines and thiophenes in hairy root cultures of *Beta vulgaris* and *Tagetes patula*. Treating *B. vulgaris* hairy roots with *H. lacustris* and *L. platensis* extracts significantly increased biomass to 165.3 g and 149.4 g fresh wt/L, respectively, from an initial 1.25 g fresh wt/L. Betalaine content in *H. lacustris*-treated *B. vulgaris* hairy roots increased 2.28-fold by day 15, while *L. platensis*-treated roots showed a 1.16-fold increase by day 25. For *T. patula, H. lacustris* extract did not affect growth or thiophene accumulation, but thiophene levels increased 1.2-fold by day 20 compared to controls. The study concludes that *L. platensis* extract enhanced betalaine production in *B. vulgaris* hairy roots, while *H. lacustris* extract improved betalaine and thiophene production in *B. vulgaris* and *T. patula* hairy roots, respectively [135].

Microalgae-based biopesticides have demonstrated effectiveness against bacteria, fungi, and oomycetes, although the specific active compound remains unidentified. Consequently, a key challenge for researchers is to pinpoint all the relevant target substances. Furthermore, the majority of experiments involving microalgal and macroalgal elicitors have been conducted in controlled environments such as in vitro systems, growth chambers, or greenhouses. This underscores the necessity for comprehensive and stringent field trials to validate their efficacy in real-world conditions [136]. The pursuit of sustainable solutions in contemporary agriculture has been driven by the recognition of chemical pesticides’ and fertilizers’ deleterious effects on the environment and human health. The transition away from conventional farming practices is simultaneously urgent and complex. It is imperative to examine the tissues affected by elicitor peptides and investigate the intricate interplay between these peptides and phytohormones in relation to various stress conditions, both abiotic and biotic.

### 2.8. Mycosporine-like Amino Acids (MAAs)

Mycosporine-like amino acids (MAAs) represent a distinctive group of marine-derived natural compounds known for their exceptional sun protection capabilities [137,138,139,140,141,142]. A remarkable feature of these compounds is their strong UV absorption, characterized by molar absorptivities (ε) of approximately 40,000 L × mol^−1^ × cm^−1^ [138,139]. MAAs are hydrophilic, low-molecular-weight compounds (typically <400 Da), consisting of either an aminocyclohexenone or aminocyclohexenimine core, with nitrogen or amino alcohol functional groups as substituents. The structures of some common MAAs isolated from cyanobacteria and microalgae are depicted in Figure 6. MAAs represent a viable natural option for both direct UV absorption and indirect antioxidant defense, owing to their strong photostability and non-toxic nature [141]. They provide photoprotection against UV-A and UV-B irradiation and exhibit distinctive absorption spectra characterized by a single, narrow, and intense absorption band, with peak absorbance occurring between 310 and 365 nm [143,144].

Research indicates that MAAs originate from the early stages of the shikimate pathway. However, the precise biochemical synthesis pathway and its underlying genetic basis remain largely unexplored [140]. Cyanobacteria are the oldest group of organisms known to synthesize MAAs. Understanding the biosynthetic pathway of MAAs is crucial for optimizing their production. Future research aims to enhance the expression of genes involved in MAA biosynthesis in microalgae by manipulating culture conditions, thereby enabling the co-production of various UV-protective compounds. This approach would expand UV coverage and streamline the production process [139,140,141,142].

MAAs are naturally occurring in aquatic organisms, both marine and freshwater, with microalgae serving as one of the most efficient MAA cell factories. In most species, MAAs are deposited extracellularly, often due to growth in biofilms or colonies. Emerging research suggests that MAAs may possess functions beyond their primary role. These compounds exhibit several important biological functions, including antioxidant activity through the neutralization of reactive oxygen species and serving as compatible solutes under salt stress conditions. In various biological systems, MAAs play a critical role in regulating osmotic balance in microalgae, cyanobacteria, and yeasts. Additionally, these versatile molecules contribute to cellular protection by scavenging free oxygen radicals and enhancing resistance to desiccation and high salinity. Two specific MAAs, mycosporine-glycine and porphyra-334, demonstrate potent antioxidant properties [145]. Microalgae synthesize MAAs when subjected to dehydration, heat, or cold stress. Furthermore, MAAs have been hypothesized to function as auxiliary pigments for light harvesting during photosynthesis and as internal nitrogen reservoirs [146,147]. More recently, novel mycosporine-like amino acids (MAAs), named klebsormidins A and B, have been identified in green algal species belonging to the Interfilum and Klebsormidium genera, which are known to inhabit extreme environments with high levels of ultraviolet radiation (UVR). In these genera, two additional MAAs, designated as MAA1 and MAA2, have also been discovered [148]. Furthermore, the red alga *Bostrychia scorpioides* has been reported to produce six novel MAAs, termed bostrychines, while *Catenella caespitosa* has been found to synthesize two unique MAAs, known as catenellines [149]. Notably, *Bostrychia scorpioides*, which thrives in harsh environments characterized by osmotic stress, desiccation, and intense UV exposure, has also been shown to produce a distinct group of seven MAAs called bostrycines [150].

The potential of MAAs as anticancer agents has been well demonstrated, particularly through their ability to inhibit the proliferation of neoplastic cells [137,138,139,140,141,142]. Furthermore, MAAs have demonstrated wound-healing properties in in vitro studies using HaCaT keratinocyte cells. The underlying mechanism of MAA action involves the activation of the focal adhesion kinase (FAK) and mitogen-activated protein kinase (MAPK) pathways. Upon introduction of MAA compounds, they induce FAK phosphorylation, which subsequently leads to the activation of MAPK extracellular signal-regulated kinase [151].

The development of advanced bioprocessing and efficient downstream processing strategies is essential for optimizing the production of MAAs from microalgae. Further research is needed to identify and understand all the key factors influencing MAA production. Bioscreening microalgae for the distribution and quantification of MAAs is a crucial aspect of microalgal biotechnology. Additionally, extraction processes must be refined by focusing on critical variables, such as temperature, duration of sonication or other extraction techniques, and the type and quantity of solvent used. These parameters are pivotal in maximizing extraction efficiency and yield [152].

### 2.9. Antivirals

Looking back at recent decades, pathogenic viruses have been a constant cause of the most severe epidemics that humans have faced. The most recent outbreaks, caused by human immunodeficiency virus (HIV), Severe Acute Respiratory Syndrome Coronavirus (SARS-CoV), the H1N1 influenza virus, Middle East Respiratory Syndrome Coronavirus (MERS-CoV), the Ebola virus, and Severe Acute Respiratory Syndrome Coronavirus-2 (SARS-CoV-2), left a severe impact on our society. In the following reviews, one can find an excellent overview of the current status regarding viral pandemics [153,154,155].

Algal compounds such as lectins and carbohydrates inhibit viral activity by forming unique bonds through selective, non-covalent interactions with viral proteins (Figure 7A) [156,157,158,159]. Carbohydrates, such as polysaccharides, can prevent infection in a number of ways [156,157,158]. Polysaccharides have been found to induce the host’s immune response to viral infection by increasing the synthesis of interferons, enhancing the proliferation of lymphocytes, and increasing the effectiveness of macrophages and NK cells (Figure 7B). Due to their negative charge, some polysaccharides can intervene in the adsorption of viruses by neutralizing their respective positive charge (Figure 7C). Polysaccharides and lectins can prevent viral infections in another way, which involves their ability to bind to either viral or host proteins (Figure 7D,F). When they bind to viral envelope proteins, they effectively inhibit the interaction between the host receptor and the viral surface protein. In addition, by binding to the host’s receptor, they occupy the binding site, which further prevents the interaction between the host and the virus. Sulfated marine polysaccharides, in particular, can impede virus internalization and uncoating by hindering the conformational changes in virus particles (Figure 7E).

Lectins are intriguing non-immune proteins that exhibit reversible selectivity for certain glycan moieties [154,155,156]. These proteins are widely distributed and identifiable in all living things, including viruses and animals. Various animal and terrestrial plant lectins have been refined and described, and a portion of these are being offered for sale for use in scientific and medical contexts, such as blood type analysis, cancer diagnosis, carbohydrate profiling, karyotyping, and evaluating patient immunocompetence [159]. Algae are a significant source of lectins, as evidenced by recent studies that described the purification, characterization, and classification of hemagglutinins from various algae species (Rhodophyta and Chlorophyta) with distinct molecular structures, biological activities, and non-promising sugar-binding specificity. Recent research has also divided lectins from algae and cyanobacteria into three primary groups: lectins specific to high mannose (HM), lectins specific to N-glycan, and lectins that exhibit specificity towards both based on their ability to bind carbohydrates [160,161,162]. The lectins purified from blue-green algae, *Microcystis aeruginosa* PCC7806 (Microvirin or MVN), *Microcystis viridis* NIES-102 (MVL), and *Nostoc ellipsosporum* (Cyanovirin-N or CV-N) (Figure 8) were among the lectins that demonstrated specificity towards the HM class [163,164,165]. The distinct carbohydrate specificity of marine algal and cyanobacterial lectins, which gives them their antiviral activity, has drawn the interest of researchers. According to reports, these lectins work against viruses by interacting with viral high-mannose glycan, which is added to the viral envelope proteins as a post-translational modification and serves as a common entry point for viruses with half-maximum effective concentrations (EC50) in the nanomolar range [165,166,167,168].

A novel lectin isolated from *Halimeda renschii* named HRL40 showed high in vitro inhibition against infection by influenza A virus [169]. A screening of 12 lectins showed that ESA-2, a lectin from the red alga *Eucheuma serra*, showed high viral inhibition against many influenza A strains [162]. Both of these lectins showed specific binding to highmannose *N*-glycans (HM-glycans) and inhibited infection with half-maximal effective doses (ED_50_) at picomolar to low nanomolar concentrations. Two lectins isolated from the species *Nostoc muscorum* also showed potent antiviral activity against HCV [170]. *Nostoc muscorum* lectin-1 (NML-1) binds preferentially to complex glycoproteins and employs its antiviral activity at the time of viral entry. *Nostoc muscorum* lectin-2 (NML-2), the second lectin analyzed, is specific to mannose polymers and inhibits viral activity post viral entry. Recently, an oscillatorial lectin isolated from the species *Oscillatoria acuminate* demonstrated virucidal qualities against herpes simplex virus type-1 (HSV-1) [171]. Another newly characterized lectin from the cyanobacterium *Lyngbya confervoides* also showed high levels of inhibitory properties against HSV-1 [172].

Vero cells pre-treated with pressurized liquid extraction (PLE) extracts from *Haematococcus lacustris* and *Dunaliella salina* demonstrated approximately 85% and 65% inhibition of infection, respectively, against herpes simplex virus type-1 (HSV-1) [173]. The ability to inhibit viral infections is believed to correlate with the polysaccharide content of the PLE extracts. In the case of PLE extracts from *H. lacustris*, short-chain fatty acids are the compounds believed to inhibit infection, while in the case of *D. salina* extracts, β-ionone, neophytadiene, phytol, palmitic acid, and α-linolenic acid are the active compounds related to viral infection inhibition. It is worth noting that cells treated with these specific PLE extracts post infection, or simultaneously with infection, did not show any protection against HSV-1. In another example, crude extracts from six microalgal species have demonstrated antiviral activity against cyprinid herpesvirus 3 (CyHV-3) [174]. Specifically, ethanolic extracts obtained via accelerated solvent extraction (ASE) from *Limnospira platensis*, *Chlamydomonas reinhardtii*, *Chlorella kessleri*, *Haematococcus lacustris*, *Nostoc punctiforme*, and *Scenedesmus obliquus* were shown to reduce viral DNA replication in vitro at non-cytotoxic concentrations. Moreover, a collection of 38 microalgal and cyanobacterial extracts was used to screen their inhibitory effects against the influenza A and B viruses [175]. The extract with the greatest inhibitory capabilities was from the Chlorellaceae family, while extracts from the Desmidiaceae and Scenedesmaceae families also showed high inhibition of viral replication. Two extracts from the green microalga *Neochloris oleoabundans* have shown protective effects against the HCoV-229E virus [176]. Cells treated prior to and during infection by the virus had highly reduced viral load, while post-infection treatment had no significant changes.

Polysaccharide extracts from *Laminaria japonica* have been shown to inhibit the replication of respiratory syncytial virus (RSV) with a demonstrated therapeutic index (TI) far greater than that of other drugs used against RSV [177]. The bioactive molecule responsible for the inhibitory properties of this aqueous extract is believed to be fucoidan, a long-chained sulfated polysaccharide. Fucoidan extracted from the species *Cladosiphon okamuranus* has also been reported to have virucidal properties against the Newcastle disease virus (NDV), one of the most severe pathogenic agents affecting birds [178]. Intriguingly, fucoidan showed better antiviral properties when compared to ribavirin, a drug used to combat NDV, due to its greatly lower toxicity. Another microalga with a sulfated polysaccharide capable of inhibiting viral infection is *Gyrodinium impudium*. The polysaccharide p-KG03, derived from *Gyrodinium impudicum*, has been identified as a specific entry inhibitor of influenza A virus, while also exhibiting non-specific inhibitory activity against influenza B virus entry [179]. It is proposed that p-KG03 interferes with multiple stages of the viral life cycle, including virus attachment, internalization into host cells, and possibly early replication events.

### 2.10. Analytical and Industrial Enzymes

Enzyme biotechnology has a positive impact on societal challenges and contributes to sustainable development in various industries, especially food, cosmetics, textiles, and feed [180,181,182,183,184,185]. Currently, the enzyme biotechnology industry primarily relies on microbial and animal enzymes, but consumer demand for alternative sustainable sources is growing due to the negative perception of animal-derived products. The development of enzymes from green and sustainable sources, like microalgae, is crucial for meeting this demand [180]. Enzymes derived from algae, such as hydrolases, cellulases, lipases, and pectinases, hold significant promise for industrial and biotechnological applications due to their diverse functionalities and sustainable production methods. Microalgae, including both prokaryotic cyanobacteria and eukaryotic photosynthetic microorganisms, are particularly advantageous as enzyme biofactories because they can be cultivated in photobioreactors, allowing for high biomass production in a cost-effective manner [186,187]. These enzymes are utilized across various industries, including food, pharmaceuticals, textiles, and biofuels, due to their ability to catalyze specific reactions efficiently. For example, cellulases are crucial in degrading cellulosic materials, essential for biofuel production and waste management [188]. Similarly, pectinases, which account for a significant portion of the global enzyme market, are vital in processes such as fruit juice clarification and textile processing [189,190].

The genetic diversity of microalgae offers a rich source of novel enzymes, and advances in genomics and transcriptomics are facilitating the identification and characterization of these enzymes for industrial use [186,187]. Moreover, the marine environment, with its vast biodiversity, provides unique enzymes that can withstand extreme conditions, making them suitable for various biotechnological applications [191]. The potential of algae as a sustainable source of industrial enzymes is further enhanced by their minimal nutritional requirements and the ability to produce enzymes with unique physicochemical properties, which are often more efficient than those derived from traditional microbial sources [192,193]. As research continues to explore and optimize the production of algal enzymes, their role in industrial applications is expected to expand, offering eco-friendly and economically viable alternatives to chemical catalysts [194,195].

The diversity of microalgae ecosystems is a potential source of useful enzymes for industry [196]. The vast metabolic diversity of microalgae likely corresponds to an equally extensive catalytic diversity, much of which remains unexplored. As a result, the diverse array of microalgal species holds great promise for the discovery of novel enzymes and biocatalysts, with the potential to drive industrial biotechnology toward greater economic viability and sustainability [180]. Due to their adaptation to unique environmental conditions, microalgae serve as a valuable source of enzymes with potentially advantageous traits, including enhanced salt tolerance, hyper-thermostability, cold adaptability, and barophilicity. Additionally, they may exhibit novel chemical and stereochemical properties, further expanding their biotechnological potential [197]. Recent microalgae genomics projects reveal potential novel enzyme genes for biotechnological applications. These enzymes often have unique properties, making them useful for industrial applications [198,199]. For example, enzymes from microalgae have shown significant potential in environmental remediation, particularly in the biodegradation of high-molecular-weight polycyclic aromatic hydrocarbons (PAHs), where proteomic studies have revealed key enzymatic activities involved in pollutant degradation pathways [200]. In the bioenergy sector, microalgal enzymes such as hydrogenase and nitrogenase are being engineered to improve bio-hydrogen production, presenting a sustainable route to clean energy [201]. Additionally, specific enzymes play a central role in directing carbon allocation and carbohydrate accumulation in microalgae, which is critical for optimizing biomass for bioethanol production [202]. In the health and nutrition fields, microalgal enzymes contribute to the biosynthesis of high-value carotenoids like astaxanthin and zeaxanthin, with genome-based studies identifying hundreds of key enzymes across various algal species [203].

The cosmetics industry is expected to drive significant growth in the global industrial enzyme market over the next decade, as consumers demand high-quality, eco-friendly, and cost-effective products [204]. In cosmetics, enzymes are being explored in anti-aging applications, as well as in the treatment of certain skin conditions such as psoriasis, eczema, and acne. They have the potential to offer a natural and safe alternative to conventional treatments, which often rely on harsh chemicals or synthetic materials [205,206].

Proteases have diverse applications across multiple industries, including food processing, detergents, and cosmetics, among others [207,208,209,210]. They are used to hydrolyze proteins, improve the texture and flavor of food, and remove stains and dirt, and they have an effect on skin exfoliation and turnover of skin cells. In dermatology, proteolytic enzymes are being explored for their potential in promoting smooth wound healing and facilitating the removal of necrotic tissue from burns and injuries [211,212]. In addition to debridement, proteolytic enzymes are being evaluated for other skin-related benefits, such as improving the appearance of aging skin, reducing scarring, and promoting wound healing [207,208,209,210]. Microalgae-derived proteases offer several advantages over plant-based counterparts, including greater stability and activity across a broader range of pH and temperature conditions. These properties make them particularly well suited for cosmetic and dermatological applications. For example, Ioannou and Labrou (2022) investigated the proteolytic activity of *Limnospira platensis* lysate [213]. Proteolytic activity was utilized to develop a hydrogel formulation as an enzyme-based cosmeceutical, with potential application as a topical exfoliating agent. The incorporation of *L. platensis* extract into the hydrogel significantly enhanced the enzyme’s long-term operational stability, a crucial advantage in the development of enzyme-based products [213]. Moreover, microalgae utilize antioxidant enzymes such as superoxide dismutase, catalase, peroxidases, and glutathione reductase to mitigate oxidative stress caused by environmental factors, heavy metals, and chemical exposure. These enzymes not only play a critical role in enhancing microalgal stress tolerance and supporting therapeutic antioxidant development but are also increasingly exploited in the cosmetics industry for their ability to protect skin cells from oxidative damage, reduce inflammation, and promote anti-aging effects [214].

## 3. Conclusions

Given their environmental and economic benefits, microalgae represent a pivotal resource for sustainable production across industries. Their role in carbon sequestration and wastewater treatment further underscores their ecological significance. Microalgae-derived compounds possess considerable commercial value and are extensively applied in the food, nutraceutical, pharmaceutical, cosmetics, and agricultural sectors, driving innovation across multiple industries. Challenges such as optimizing culture conditions, scaling production, and reducing costs need to be addressed to maximize commercial potential. Continued research in genetic engineering, metabolic pathways, and advanced bioprocessing techniques are crucial in unlocking their full capabilities. Microalgae-derived innovations are expected to play a crucial role in addressing global challenges related to health, food security, and environmental sustainability.

## Figures and Tables

**Figure 1 marinedrugs-23-00222-f001:**
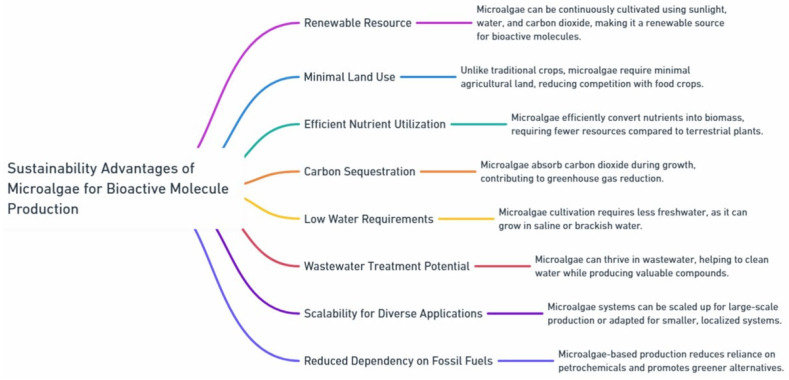
Sustainability advantages of microalgae for bioactive molecule production. Microalgae offer a renewable, eco-friendly source for bioactive compounds by utilizing sunlight, water, and CO_2_ efficiently. They require minimal land and can grow in saline or brackish water. Microalgae contribute to carbon sequestration, can treat wastewater, and offer scalable production systems for various applications. Additionally, they reduce dependence on fossil fuels, supporting sustainable and greener alternatives.

**Figure 2 marinedrugs-23-00222-f002:**
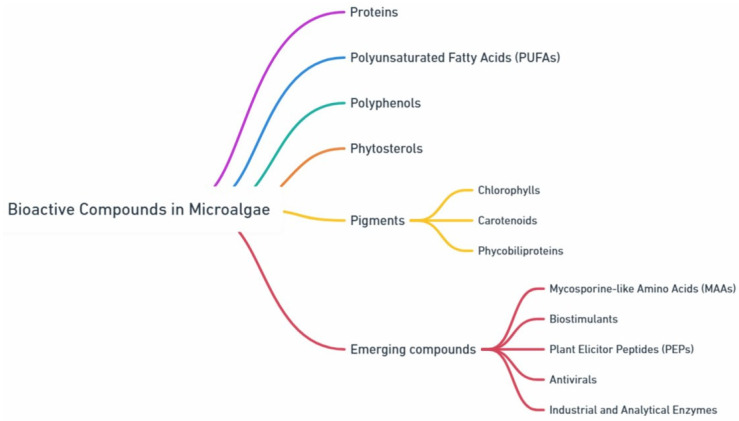
Important categories of bioactive compounds from microalgae, such as proteins, polyunsaturated fatty acids (PUFAs), polyphenols, phytosterols, pigments. Emerging compounds encompass mycosporine-like amino acids (MAAs), which act as UV-absorbing compounds, biostimulants that promote plant growth, plant elicitor peptides (PEPs) that trigger plant defenses, antivirals with potential to inhibit viral replication, and industrial and analytical enzymes that are used in manufacturing and research.

**Figure 3 marinedrugs-23-00222-f003:**
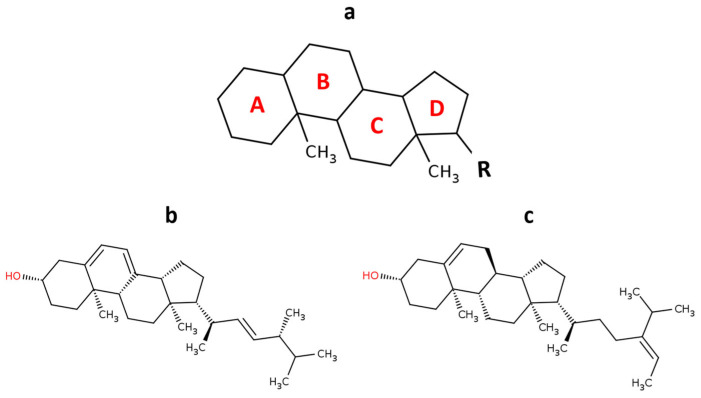
Representative chemical structures of phytosterols derived from microalgae. (**a**) The steroid skeleton. Sterols are based on the cyclopenta[a]phenanthrene carbon skeleton (rings A, B, C, and D), typically bearing methyl groups at positions C-10 and C-13, and often an alkyl substituent (R) at C-17. (**b**) Ergosterol, a sterol found in several microalgae species and known for its anticancer activity. (**c**) Fucosterol, a phytosterol commonly isolated from brown algae, associated with antioxidant, anti-inflammatory, and cholesterol-lowering effects.

**Figure 4 marinedrugs-23-00222-f004:**
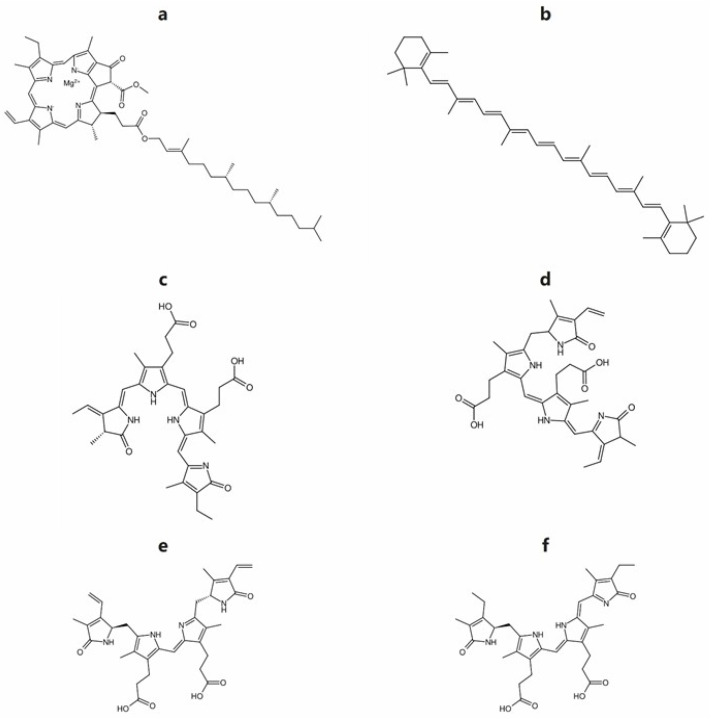
The structure of chlorophyll a (**a**), β-carotene (**b**), and different types of phycobilins: phycocyanobilin (**c**), phycoerythrobilin (**d**), phycourobilin (**e**), and phycoviolobilin (**f**).

**Figure 5 marinedrugs-23-00222-f005:**
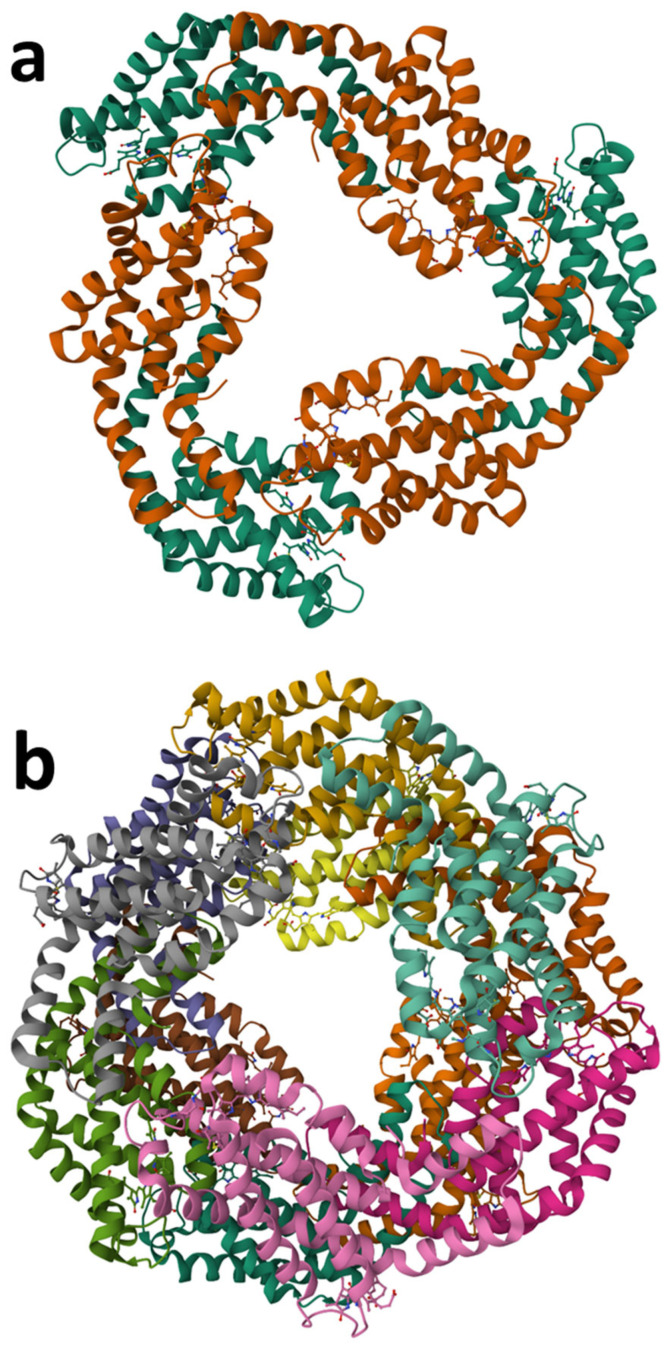
Ribbon model of the structure of PBPs. (**a**) The structure of *Thermosynechococcus elongatus* allophycocyanin (PDB code 2V8A). (**b**) The structure of *Limnospira platensis* Cphycocyanin (PDB code 1HA7). (**c**) The structure of C-phycoerythrin from marine cyanobacterium *Phormidium* sp. A09DM (PDB code 5NB3). The bound phycobilins are shown in a ball and stick representation.

**Figure 6 marinedrugs-23-00222-f006:**
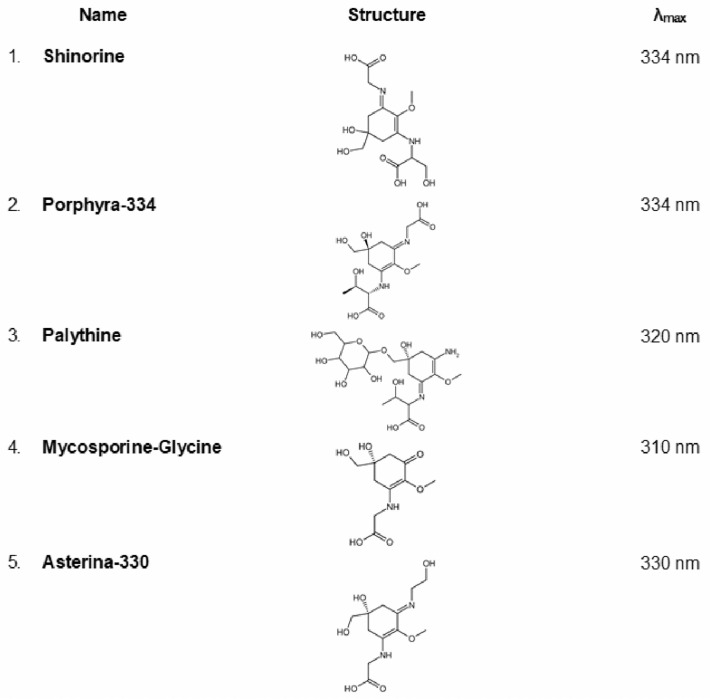
Some of the most commonly identified mycosporine-like amino acids (MAAs) in cyanobacteria and microalgae, along with their respective absorption maxima (λ_max_, determined in distilled water).

**Figure 7 marinedrugs-23-00222-f007:**
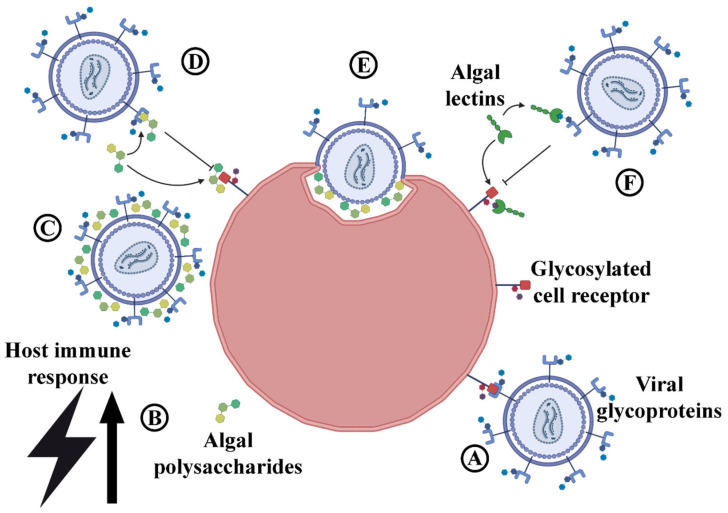
Graphical representation of antiviral algal compounds and their mechanism. (**A**) Viral envelope proteins bind to the host’s membrane receptors in the first step of infection uninterrupted, in the absence of antiviral compounds. (**B**) Algal polysaccharides have been observed to increase the host’s immune response, which in turn helps reduce the strain caused by infection. (**C**) Algal polysaccharides can also prevent infection by neutralizing their positive charge. (**D**) They can also bind to either host or viral proteins thus preventing the interaction between the two. (**E**) Algal polysaccharides can prevent viral conformational changes and inhibit adsorption. (**F**) Lectins bind in a specific manner to glycosylated proteins, either on the viral envelope or the host’s cell membrane, occupying the binding site and preventing interaction.

**Figure 8 marinedrugs-23-00222-f008:**
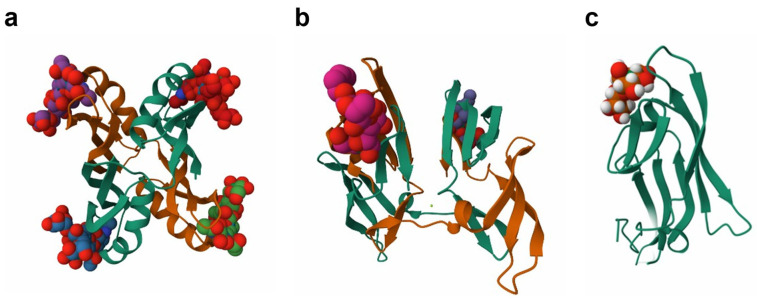
Ribbon model of the structures of antiviral lectins. (**a**) Lectin from *Microcystis aeruginosa* PCC7806 (Microvirin or MVN) (PDB code: 2YHH, crystal structure of MVN bound to mannobiose). (**b**) Lectin from *Microcystis viridis* NIES-102 (MVL) (PDB code: 1ZHS, crystal structure of MVL bound to Man3GlcNAc2). (**c**) Lectin from *Nostoc ellipsosporum* (CV-N) (PDB code: 3GXY, crystal structure of CV-N bound to a synthetic hexamannoside).

## Data Availability

Not applicable.

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
