# Peer review of "Diversity of Bioactive Compounds in Microalgae: Key Classes and Functional Applications"

_marinedrugs, 2025, doi:10.3390/md23060222_

Round 1
Reviewer 1 Report
Comments and Suggestions for Authors
1. As a general note, it is apparent that different sections were contributed by multiple authors, as there is a change in tone, voice and writing technique between sections. This is due to different personal writing styles, but in my opinion this makes for an uneven reading experience. In addition, some sections have low usage of citations (section 2.6) while others appear to cite more articles than necessary (section 2.5.3). I would suggest an overall re-edit of the manuscript with the intention of making the text more homogenous in regards to writing style and citation usage.
2. In Table 1, there may be a typo in the left column “is Limnospira platensis (previously Spirulina platensis)”. Is the “is” meant to be in there? I would also consider removing the mention of the previous naming from the table and address it in the text, for the sake of space.
3. Section 2.2 (PUFAs) needs to be expanded, as I feel this topic is slightly glossed over. For example, Spirulina is mentioned both in Tables 1 and 2, but there is no mention in section 2.2 about the contribution of PUFAs from Spirulina strains in nutraceutical applications.
4. It is unclear to me why some sections include figures, while others do not. For example, if you wish to include Figure 6, which relates to section 2.9, why not include a figure for section 2.7 (plant elicitor peptides)? The interactions of plants with their rhizospheres and potential pathogens are complex and may be better conveyed by having an accompanying figure and chemical structures of known peptides could be included for illustration.
5. There are no shown chemical structures for the section describing phytosterols (section 2.4). Perhaps one or two could be added into Figure 3 since it already has structures mentioned in multiple other sections.
6. Regarding Figure 4, it is difficult to see the ball-and-stick representation of the bound chromophores. Can this figure be enlarged or modified in order to make the small chemical structures more visible?
7. In section 2.6, it is stated “...Polysaccharides and lectins can prevent viral infections in another way, which involves their ability to bind to either viral or host proteins (Fig. 5D, F).” This should be changed to reference Figure 6D and 6F.
8. Section 2.10 has a large paragraph describing potential algal industrial and analytical enzymes without naming a single enzyme or describing a mechanism. What are the useful industrial enzymes what algae could provide? For example, hydrolase, cellulase or pectinase? Also, citations 163-170 are all other review articles, without a single original research paper in this area cited. This section would be more interesting and informative if it cited some original research on specific enzymes.
Comments on the Quality of English Language
The quality of the English is good in this manuscript, however there are some sections that contain less formal usages of English that are not usually applied in scientific articles. Please also refer to my first specific comment regarding the overall flow of the article for more points.
Author Response
- As a general note, it is apparent that different sections were contributed by multiple authors, as there is a change in tone, voice and writing technique between sections. This is due to different personal writing styles, but in my opinion this makes for an uneven reading experience. In addition, some sections have low usage of citations (section 2.6) while others appear to cite more articles than necessary (section 2.5.3). I would suggest an overall re-edit of the manuscript with the intention of making the text more homogenous in regards to writing style and citation usage.
Response: We have edited the text with the help native English speaking professors to normalize the text
- In Table 1, there may be a typo in the left column “is Limnospira platensis (previously Spirulina platensis)”. Is the “is” meant to be in there? I would also consider removing the mention of the previous naming from the table and address it in the text, for the sake of space.
Response: Thank you for pointing this out. We agree that the word “is” in the left column of Table 1 appears to be a typographical error and have removed it. We also appreciate your suggestion regarding the naming convention. To improve clarity and conserve space in the table, we have removed the mention of the previous name “Spirulina platensis” from Table 1 and now address this nomenclatural update in the main text instead.
- Section 2.2 (PUFAs) needs to be expanded, as I feel this topic is slightly glossed over. For example, Spirulinais mentioned both in Tables 1 and 2, but there is no mention in section 2.2 about the contribution of PUFAs from Spirulinastrains in nutraceutical applications.
Response: We appreciate your observation regarding the limited coverage of Spirulina's role in Section 2.2. In response, we have expanded the section to include a concise but meaningful discussion on the contribution of Spirulina (now classified as Limnospira platensis) to PUFA production, particularly highlighting its relevance as a source of γ-linolenic acid (GLA) and its applications in nutraceutical products. This addition aligns with its inclusion in Tables 1 and 2 and provides a more balanced overview of PUFA-producing microalgae. We have also ensured that the revised text maintains an even emphasis across various species for improved clarity and flow.
- It is unclear to me why some sections include figures, while others do not. For example, if you wish to include Figure 6, which relates to section 2.9, why not include a figure for section 2.7 (plant elicitor peptides)? The interactions of plants with their rhizospheres and potential pathogens are complex and may be better conveyed by having an accompanying figure and chemical structures of known peptides could be included for illustration.
Response: Thank you for your helpful suggestion. We agree that the use of figures can significantly enhance the reader’s understanding, especially in sections covering complex biological interactions such as Section 2.7. In this case, however, we opted not to include an additional figure to maintain a balanced visual presentation across the sections and avoid potential redundancy with the detailed textual explanations already provided. The mechanisms involving plant elicitor peptides, while important, are still under active investigation, and at present there is limited consensus on standardized models or representative structures for inclusion. We have aimed to provide a comprehensive narrative that conveys the complexity of PEP interactions and their role in plant defense, while maintaining consistency in the overall layout and length of the manuscript. We hope this explanation is acceptable and that the detailed content in Section 2.7 provides sufficient clarity for readers.
- There are no shown chemical structures for the section describing phytosterols (section 2.4). Perhaps one or two could be added into Figure 3 since it already has structures mentioned in multiple other sections.
Response: Thank you for your helpful suggestion. We agree that including representative chemical structures of phytosterols would enhance the clarity of Section 2.4. To maintain consistency and avoid overloading Figure 3, which is dedicated to microalgal pigments (Section 2.5), we have instead created a new figure (now added as Figure 3) to illustrate the chemical general structure of sterols as well as the structures of ergosterol and fucosterol, which are discussed in Section 2.4. We believe this approach provides better clarity and structural alignment within the manuscript.
- Regarding Figure 4, it is difficult to see the ball-and-stick representation of the bound chromophores. Can this figure be enlarged or modified in order to make the small chemical structures more visible?
Response: We understand the concern regarding the visibility of the bound chromophores in the ball-and-stick representation in Figure 4. In response, we have enlarged the figure and adjusted the resolution to improve the clarity of the chemical structures. We have also slightly modified the layout to ensure that the bound chromophores are more distinct and easier to interpret. We hope this revision improves the readability and visual quality of the figure.
- 7. In section 2.6, it is stated “...Polysaccharides and lectins can prevent viral infections in another way, which involves their ability to bind to either viral or host proteins (Fig. 5D, F).” This should be changed to reference Figure 6D and 6F.
Response: Thank you for pointing out the incorrect figure reference. We have corrected the citation in Section 2.9 to reference Figure 6D and 6F instead of Figure 5D and 5F, ensuring consistency and accuracy in the manuscript.
- Section 2.10 has a large paragraph describing potential algal industrial and analytical enzymes without naming a single enzyme or describing a mechanism. What are the useful industrial enzymes what algae could provide? For example, hydrolase, cellulase or pectinase? Also, citations 163-170 are all other review articles, without a single original research paper in this area cited. This section would be more interesting and informative if it cited some original research on specific enzymes.
Response: We appreciate your suggestion to strengthen Section 2.10 by naming specific industrially relevant enzymes and including original research. In response, we have revised the section to mention key enzymes such as hydrolases, cellulases, lipases, and pectinases, which have demonstrated potential for industrial and biotechnological applications. We have also added brief descriptions of their functions and relevance.
Furthermore, we have updated the references to include original research articles that report on the isolation, characterization, or application of these enzymes from various algal sources. These additions provide more depth and specificity to the section, aligning it with the rest of the manuscript and enhancing its value to readers interested in algal biotechnology.
Reviewer 2 Report
Comments and Suggestions for Authors
Initially, the manuscript's substantial number of contributors, specifically 28 authors, merits attention.
The purpose of the review should be stated, as well as the system used to select papers and their citations. What keywords were used for the search and the selection method and databases used. It is not stated in the manuscript whether the search had a reference period for the search of the articles.
A multitude of article reviews within the scientific literature address the compounds present in microalgae, encompassing recent reviews that delve into their applications.
A significant number of inaccuracies in name indications and in the text have been identified. The text of the manuscript requires extensive revision, as its current state is characterized by substandard writing and organization, hindering the reader's comprehension and interrupting the coherence of the different sections and paragraphs.
The title refers to microalgae; however, it fails to provide any indication as to the specific species of microalgae under consideration. It is imperative that the authors incorporate a paragraph that elucidates the rationale behind the selection of particular microalgae species. Numerous reviews in scientific literature addressing the applications of microalgae compounds have previously documented this information.
2.High added-value compounds from microalgae - In this section, there are groups of compound classes with very little description compared to other described extensions.
2. High added-value compounds from microalgae – In this chapter, a selection of compound classes from microalgae was made, but not a selection of applications, as indicated in the title. The authors could subdivide the compound class paragraph adding functional applications (e.g., 2.1 Proteins into a sub-section of 2.1.1 Functional applications).
A more thorough and revised review is necessary because the novelty of the subject matter, as compared to the extant scientific literature, and its potential interest to researchers is not clear. The role of the numerous co-authors is also unclear.
Author Response
Initially, the manuscript's substantial number of contributors, specifically 28 authors, merits attention.
Response The invited review article, titled “Diversity of Bioactive Compounds in Microalgae: Key Classes and Functional Applications”, has been authored by members of the AlgaeNet4AV consortium. AlgaeNet4AV (http://algaenet4av.eu/) is a research and innovation program funded by the European Union under the Marie SkÅ‚odowska-Curie Actions Staff Exchanges scheme (HE MSCA SE). The consortium, composed by twelve industrial and academic partners, aims to explore microalgal diversity for the discovery and development of novel antiviral compounds. All individuals listed as authors have contributed to the selection and analysis of relevant literature, as well as to the writing, editing and proofreading of the review article.
The purpose of the review should be stated, as well as the system used to select papers and their citations. What keywords were used for the search and the selection method and databases used. It is not stated in the manuscript whether the search had a reference period for the search of the articles.
Response: We include the following statement “This review highlights the increasing prominence of microalgae derived compounds across agricultural and pharmaceutical applications, underscoring their versatile potential in meeting the needs of diverse industries.”
A multitude of article reviews within the scientific literature address the compounds present in microalgae, encompassing recent reviews that delve into their applications.
A significant number of inaccuracies in name indications and in the text have been identified. The text of the manuscript requires extensive revision, as its current state is characterized by substandard writing and organization, hindering the reader's comprehension and interrupting the coherence of the different sections and paragraphs.
Response: We have edited all names as suggested
The title refers to microalgae; however, it fails to provide any indication as to the specific species of microalgae under consideration. It is imperative that the authors incorporate a paragraph that elucidates the rationale behind the selection of particular microalgae species. Numerous reviews in scientific literature addressing the applications of microalgae compounds have previously documented this information.
Response: Thank you for your comment however the review is based on the bioactive compounds and we then consider which microalgae produce those, we do not dicsibe microalgae species and which bioactive compound each species produces
2.High added-value compounds from microalgae - In this section, there are groups of compound classes with very little description compared to other described extensions.
Response: We have included more details in the relative sections
- High added-value compounds from microalgae – In this chapter, a selection of compound classes from microalgae was made, but not a selection of applications, as indicated in the title. The authors could subdivide the compound class paragraph adding functional applications (e.g., 2.1 Proteins into a sub-section of 2.1.1 Functional applications).
A more thorough and revised review is necessary because the novelty of the subject matter, as compared to the extant scientific literature, and its potential interest to researchers is not clear. The role of the numerous co-authors is also unclear.
Response: We have edited the review and added more literature
Reviewer 3 Report
Comments and Suggestions for Authors
REVIEW OF THE ARTICLE BY MASLIN OSATHANUNKUL ET AL. ENTITLED ‘DIVERSITY OF BIOACTIVE COMPOUNDS IN MICROALGAE: KEY CLASSES AND FUNCTIONAL APPLICATIONS’
Osathanunkul et al. summarise current knowledge on different groups of bioactive compounds from microalgae, including polysaccharides, pigments, mycosporine-like amino acids, lectins, etc. They describe their origins with examples of algal species, and their role in human life. The review is timely, many recent references are used. It is in the scope of the journal. In general, the text is well-written. At the same time there are some drawbacks. The text makes an impression that different sections were written by different people and has not been integrated into a whole text. Although the authors cite many recent works, in many cases they use old algal names. Curiously, in different sections different synonyms are used for the same alga. I suggest revision of the review in accordance with points below.
GENERAL COMMENTS
-Use current correct algal names (according to AlgaeBase, https://www.algaebase.org/): Dunaliella bardawil → Dunaliella salina, Haematococcus pluvialis → Haematococcus lacustris, Chlorella zofingiensis → Chromochloris zofingiensis, Chlorella photothecoides → Auxenochlorella protothecoides, Spirulina platensis → Arthrospira platensis, Chlorella fusca → Desmodesmus abundans, Oscillatoria acuminate → Oxynema acuminatum, Nostoc muscorum → Desmonostoc muscorum, Chlorella kessleri → Parachlorella kessleri, Scenedesmus obliquus → Tetradesmus obliquus, Neochloris oleoabundans → Ettlia oleoabundans, Gyrodinium impudium → Gymnodinium impudicum.
INTRODUCTION
-1st paragraph: a lot of information without references.
-Figure 1. Yellow line: Low Freshwater Requirements?
-Ad reference to EABA to the reference list.
-3rd paragraph: dot is missing in the end.
-4th paragraph: a lot of information is without references.
-Table 1: ‘previously’ should not be italicised.
-Footnote is missing.
SECTION 2.1
-meat, dairy, and eggs are not proteins. They are protein-rich products.
-Algal genera and species should be italicised: Chlorella, Spirulina, Limnospira.
SECTION 2.2
-‘Monodus subterraneus’, ‘Pavlova girans’ - wrong spelling.
-All species names should be italicised.
-All species names must be italicised.
-You should mention Lobasphaera incisa as one of the main algal sources of arachidonic acid.
SECTION 2.3
-‘coronavirus infection’ - which coronavirus? It is a large group of viruses.
-HIV - abbreviation should be described at the first mention.
SECTION 2.4
-Chlorophyceae, Phaeophyceae and Rhodophyceae are classes of algae, not families.
-Comma after ‘For example’, ‘Furthermore’.
-P. lutheri - write in full at the first mention.
-double ‘in’ - a typing error.
-β-carotenoid should be β-carotene.
-Schizochytrium is not an alga and, therefore, is out of scope of the review.
SECTION 2.5
-‘energy absorbers in the photosynthetic system of microalgae’ - it is wrong. Secondary carotenoids are not a part of photosynthetic apparatus.
-‘beta-carotene’ should be ‘β-carotene’.
-If you indicate absorbance maxima, you should also indicate solvent in which it was registered.
-‘Chlorella’ - should be italicised.
-In one-two sentences, please, figure out the diversity and distribution of algal chlorophylls (see e.g. 10.3390/md9061101).
-‘. and ’ - a typing error.
-Your ref. [91] is about astaxanthin-producing astaxanthin. Production of astaxanthin with β-carotene: 10.3390/biology10070643.
-Revise algal names (see general comments).
-Expand the current knowledge on carotenoid-accumulating algae: 10.3390/md21020108.
-It is nicessary to mention fucoxanthin, although it is not a secondary carotenoid: 10.3390/md20040222.
-‘Another area of application is their medical properties.’ - revise the sentence. Properties cannot be applications.
-Please, figure out distribution of phycobilins across different algal groups.
-‘Arthrospira’ should be italicised.
-‘cell cycle of cancer cells’ - please, rephrase.
SECTION 2.6
-‘Plant development is promoted when microalgae, cyanobacteria or their formulation like (biomass, extracts, hydrolysates)’ - cannot understand. Revise.
-‘Porphirydium’ - should be italicised.
-Please, pay more attention to the chemical nature of these stimulants.
SECTION 2.7
-Why do you separate elicitors from other bioactive molecules (previous subsection)?
-1st paragraph: a lot of information without references.
-‘This phenomenon has … flavonoid and phenolic compounds’ - why do you attributes this to immune response? As a rule, the response against pathogens is accompanied to a high ROS level, whereas you say about the expression of SOD, CAT and APX, which reduce their level.
-‘various microalgae species, including Chlorella and Dunaliella (members of Chlorophyta), as well as Porphyridium (belonging to Rhodophyta)’ - these are genera of algae (not species). Genera names should be italicised.
-‘Additionally, green microalgae have the ability to produce lactic acid …’ - why not in the previous section?
-‘spp.’ - should not be italicised.
-Phyla names should not be italicised.
-Check algal names (see general comments).
- ‘Rao et al. (2001), studied’ - remove comma.
-‘Rao et al. (2001), studied …’ - H. lacustris and A. platensis are not elicitors, they alre algal species.
-‘B. vulgaris’ - should be italicised.
-last paragraph: a lot of information without references.
SECTION 2.8
-Figure 5: in the legend, describe what the 3rd column means.
-‘unexplored.’ - remove the dot.
-‘the most ancient organisms’ - the most ancient group of organisms.
-‘to enhance MAA expression’ - there is no MMA expression, there is an expression of genes.
As this is a review of recent literature, I would focus on newly discovered MAAs instead of old ones. Please, mention with corresponding references klebsormidins, coelastrins, bostrychines, and catenellines.
-Last paragraph: the abbreviation MAA has been introduced earlier.
-If you indicate absorbance maxima, you should also indicate solvent in which it was registered.
SECTION 2.9
-‘Human Immunodeficiency Virus (HIV)’ - the abbreviation should be introduced earlier.
-‘Algae compounds’ - ‘Algal compounds’.
-‘non-catalytic selectivity’ - what is it?
- ‘nan-molar’ - nanomolar?
-If you introduce an abbreviation (HM), use it through the text.
-Lyngabya confervoides - wrong spelling.
-Halimeda renchii - wrong spelling.
-‘sufated polysaccharide’ - check spelling.
-Please revise algal names in accordance with the recent taxonomy (see general comments).
Comments on the Quality of English Language
In general, the text is readable, but there are some mistakes. I tried to fix them (see comments to the authors). The text should be double checked at the stage of English correction in the case of acceptance.
Author Response
Osathanunkul et al. summarise current knowledge on different groups of bioactive compounds from microalgae, including polysaccharides, pigments, mycosporine-like amino acids, lectins, etc. They describe their origins with examples of algal species, and their role in human life. The review is timely, many recent references are used. It is in the scope of the journal. In general, the text is well-written. At the same time there are some drawbacks. The text makes an impression that different sections were written by different people and has not been integrated into a whole text. Although the authors cite many recent works, in many cases they use old algal names. Curiously, in different sections different synonyms are used for the same alga. I suggest revision of the review in accordance with points below.
Response: thank you for the suggestion we have now used the proposed names in all text furthermore all the text has been edited from 3 different people to unify the text
GENERAL COMMENTS
-Use current correct algal names (according to AlgaeBase, https://www.algaebase.org/): Dunaliella bardawil → Dunaliella salina, Haematococcus pluvialis → Haematococcus lacustris, Chlorella zofingiensis → Chromochloris zofingiensis, Chlorella photothecoides → Auxenochlorella protothecoides, Spirulina platensis → Arthrospira platensis, Chlorella fusca → Desmodesmus abundans, Oscillatoria acuminate → Oxynema acuminatum, Nostoc muscorum → Desmonostoc muscorum, Chlorella kessleri → Parachlorella kessleri, Scenedesmus obliquus → Tetradesmus obliquus, Neochloris oleoabundans → Ettlia oleoabundans, Gyrodinium impudium → Gymnodinium impudicum.
Response: thank you for the suggestion we have now used the proposed names
INTRODUCTION
-1st paragraph: a lot of information without references.
Response: Relevant references were added.
-Figure 1. Yellow line: Low Freshwater Requirements?
Response: was omitted
-Ad reference to EABA to the reference list.
Response: The reference was moved in the reference list
-3rd paragraph: dot is missing in the end.
Response: Added.
-4th paragraph: a lot of information is without references.
Response: Added.
-Table 1: ‘previously’ should not be italicised.
Response: Removed as suggested by another reviewer.
-Footnote is missing.
Response: Added.
SECTION 2.1
-meat, dairy, and eggs are not proteins. They are protein-rich products.
Response: Revised.
-Algal genera and species should be italicised: Chlorella, Spirulina, Limnospira.
Response: Edited.
SECTION 2.2
-‘Monodus subterraneus’, ‘Pavlova girans’ - wrong spelling.
Response: Edited.
-All species names should be italicised.
Response: Edited.
-All species names must be italicised.
Response: Edited.
-You should mention Lobasphaera incisa as one of the main algal sources of arachidonic acid.
Response: This information and a reference were added in the text.
SECTION 2.3
-‘coronavirus infection’ - which coronavirus? It is a large group of viruses.
Response: “SARS-CoV-2” was added.
-HIV - abbreviation should be described at the first mention.
Response: Edited.
SECTION 2.4
-Chlorophyceae, Phaeophyceae and Rhodophyceae are classes of algae, not families.
Response: Edited.
-Comma after ‘For example’, ‘Furthermore’.
Response: Edited.
-P. lutheri - write in full at the first mention.
Response: Edited.
-double ‘in’ - a typing error.
Response: Edited.
-β-carotenoid should be β-carotene.
Response: Edited.
-Schizochytrium is not an alga and, therefore, is out of scope of the review.
Response: was removed
SECTION 2.5
-‘energy absorbers in the photosynthetic system of microalgae’ - it is wrong. Secondary carotenoids are not a part of photosynthetic apparatus.
Response: This sentence was removed from the text.
-‘beta-carotene’ should be ‘β-carotene’.
Response: Edited.
-If you indicate absorbance maxima, you should also indicate solvent in which it was registered.
Response: The information was added in the text.
-‘Chlorella’ - should be italicised.
Response: Edited.
-In one-two sentences, please, figure out the diversity and distribution of algal chlorophylls (see e.g. 10.3390/md9061101).
Response: Edited.
-‘. and ’ - a typing error.
Response: Edited.
-Your ref. [91] is about astaxanthin-producing astaxanthin. Production of astaxanthin with β-carotene: 10.3390/biology10070643.
Response: Edited.
-Revise algal names (see general comments).
Response: We have revised the names as in the comments .
-Expand the current knowledge on carotenoid-accumulating algae: 10.3390/md21020108.
Response: We have edited the section and added more information according to the suggestion
-It is nicessary to mention fucoxanthin, although it is not a secondary carotenoid: 10.3390/md20040222.
Response: We now mention fucoxanthin as suggested
-‘Another area of application is their medical properties.’ - revise the sentence. Properties cannot be applications.
Response: Edited.
-Please, figure out distribution of phycobilins across different algal groups.
Response: We have revised the Distribution of phycobilins in different algal
-‘Arthrospira’ should be italicised.
Response: Edited.
-‘cell cycle of cancer cells’ - please, rephrase.
Response: We have revised the phrase “cell cycle of cancer cells” to “induce cell cycle arrest in cancer cells” to improve clarity and scientific accuracy.
PBPs have been shown to induce cell cycle arrest in cancer cells associated with liver cancer [105], breast cancer [106], leukemia [107], lung cancer [108], and bone marrow cancer [109].
SECTION 2.6
-‘Plant development is promoted when microalgae, cyanobacteria or their formulation like (biomass, extracts, hydrolysates)’ - cannot understand. Revise.
Response: We have revised the sentence in Section 2.6 for clarity. The updated version now reads: “Plant development can be enhanced by the application of microalgae, cyanobacteria, or their formulations, such as biomass, extracts, or hydrolysates. The effectiveness of these biostimulants depends on the species of microalgae and the origin of the formulation used.”
-‘Porphirydium’ - should be italicised.
Response: Edited.
-Please, pay more attention to the chemical nature of these stimulants.
Response: Edited.
SECTION 2.7
-Why do you separate elicitors from other bioactive molecules (previous subsection)?
Response: We acknowledge your observation regarding the separation of elicitors from other bioactive molecules discussed in previous subsections. The structure of Section 2 was intentionally designed to differentiate microalgal compounds based on their primary functions and modes of action.
Subsections 2.1 to 2.5 describe specific classes of bioactive molecules (e.g., proteins, PUFAs, pigments), while 2.6 (Biostimulants) highlights general formulations that enhance plant growth. In contrast, Section 2.7 focuses specifically on Plant Elicitor Peptides (PEPs) and other microalgae-derived compounds that trigger plant immune responses, such as systemic acquired resistance (SAR) and induced systemic resistance (ISR).
These elicitors differ mechanistically and functionally from compounds that act as direct nutrients or growth promoters. They serve a specialized role by priming plants against biotic and abiotic stress, which makes them a unique and emerging tool in sustainable agriculture. To emphasize their significance and distinct biological function, we have placed them in a separate subsection. This organization allows for a clearer narrative and better highlights the multifunctional potential of microalgal compounds.
-1st paragraph: a lot of information without references.
Response: Added.
-‘This phenomenon has … flavonoid and phenolic compounds’ - why do you attributes this to immune response? As a rule, the response against pathogens is accompanied to a high ROS level, whereas you say about the expression of SOD, CAT and APX, which reduce their level.
Response: The application of algae in the root system is seen by the plants as a message of infection so it stars to overexpress the antioxidant enzymes (accordint to the literature), as infection among other effect results in oxidative stress.
-‘various microalgae species, including Chlorella and Dunaliella (members of Chlorophyta), as well as Porphyridium (belonging to Rhodophyta)’ - these are genera of algae (not species). Genera names should be italicised.
Response: Edited.
-‘Additionally, green microalgae have the ability to produce lactic acid …’ - why not in the previous section?
Response: We agree with the reviewer comment and the paragraph has been moved to the previous section
-‘spp.’ - should not be italicised.
Response: Edited.
-Phyla names should not be italicised.
Response: Edited.
-Check algal names (see general comments).
Response: Edited.
- ‘Rao et al. (2001), studied’ - remove comma.
Response: Edited.
-‘Rao et al. (2001), studied …’ - H. lacustris and A. platensis are not elicitors, they alre algal species.
Response: We acknowledge the inaccuracy in our original wording. In the revised manuscript, we have clarified that Haematococcus pluvialis and Spirulina platensis are microalgal species that were used as elicitor treatments in the referenced study. The sentence has been revised as follows:
"Rao et al. (2001) studied the effects of two microalgal species, Haematococcus pluvialis and Spirulina platensis, used as elicitor treatments, on the accumulation of betalaines and thiophenes in hairy root cultures of Beta vulgaris and Tagetes patula."
-‘B. vulgaris’ - should be italicised.
Response: Edited.
-last paragraph: a lot of information without references.
Response: Added.
SECTION 2.8
-Figure 5: in the legend, describe what the 3rd column means.
Response: added
-‘unexplored.’ - remove the dot.
Response: Edited.
-‘the most ancient organisms’ - the most ancient group of organisms.
Response: Edited.
-‘to enhance MAA expression’ - there is no MMA expression, there is an expression of genes.
Response: Thank you for pointing this out. We agree that “MAA expression” was imprecise, as it is the expression of genes involved in MAA biosynthesis that is being modulated. The sentence has been revised accordingly to clarify this point.
As this is a review of recent literature, I would focus on newly discovered MAAs instead of old ones. Please, mention with corresponding references klebsormidins, coelastrins, bostrychines, and catenellines.
Response: We have added the relative text and literature
-Last paragraph: the abbreviation MAA has been introduced earlier.
Response: Edited.
-If you indicate absorbance maxima, you should also indicate solvent in which it was registered.
Response: Edited.
SECTION 2.9
-‘Human Immunodeficiency Virus (HIV)’ - the abbreviation should be introduced earlier.
Response: Edited.
-‘Algae compounds’ - ‘Algal compounds’.
Response: Edited.
-‘non-catalytic selectivity’ - what is it?
Response: The term non-catalytic refers to the fact that lectins bind sugars without inducing a chemical reaction. However, it was removed from the text to improve clarity and avoid potential confusion.
- ‘nan-molar’ - nanomolar?
Response: Edited.
-If you introduce an abbreviation (HM), use it through the text.
Response: Edited.
-Lyngabya confervoides - wrong spelling.
Response: Edited.
-Halimeda renchii - wrong spelling.
Response: Edited.
-‘sufated polysaccharide’ - check spelling.
Response: Edited.
-Please revise algal names in accordance with the recent taxonomy (see general comments).
Response: We have used the proposed names throughout the text
Comments on the Quality of English Language
In general, the text is readable, but there are some mistakes. I tried to fix them (see comments to the authors). The text should be double checked at the stage of English correction in the case of acceptance.
Answer English speaking collaborators have edited the manuscript
Round 2
Reviewer 2 Report
Comments and Suggestions for Authors
The authors responded to most of the comments and revised the manuscript accordingly. The present manuscript is now better and deserves to be published in the journal.
Author Response
Thank you for your comments
Reviewer 3 Report
Comments and Suggestions for Authors
The authors have imporved the text. However there are some drawbacks. Particularly, You use both names simultaneously: Haematococcus pluvialis vs Haematococcus lacustris, Arthrospira platensis vs Spirulina platensis.
According to the suggestions the authors added some information, e.g. regarding diversity of chlorophylls, carotenoids and phycobilins. However the most of references are to old. I suggest adding most recent articles (last 3-5 years), e.g. [90-115]. I provided some recommendations in my review.
Author Response
Thank you very much for the constructive comments. The following amendments have been made in accordance with the reviewer’s suggestions:
1) The authors have imporved the text. However there are some drawbacks. Particularly, You use both names simultaneously: Haematococcus pluvialis vs Haematococcus lacustris, Arthrospira platensis vs Spirulina platensis.
Reply: All microalgae names have been updated as per the reviewer’s recommendation.
2) According to the suggestions the authors added some information, e.g. regarding diversity of chlorophylls, carotenoids and phycobilins. However the most of references are to old. I suggest adding most recent articles (last 3-5 years), e.g. [90-115]. I provided some recommendations in my review.
Reply: The previously cited references [90–115] have been replaced with more recent literature. The majority of the updated references are from studies published between 2023 and 2025
Round 3
Reviewer 3 Report
Comments and Suggestions for Authors
-